# Oral Health-Related Quality of Life in Rare Disorders of Congenital Facial Weakness

**DOI:** 10.3390/ijerph21050615

**Published:** 2024-05-13

**Authors:** Denise K. Liberton, Konstantinia Almpani, Rashmi Mishra, Carol Bassim, Carol Van Ryzin, Bryn D. Webb, Ethylin Wang Jabs, Elizabeth C. Engle, Francis S. Collins, Irini Manoli, Janice S. Lee

**Affiliations:** 1Craniofacial Anomalies and Regeneration Section, National Institute of Dental and Craniofacial Research, National Institutes of Health, Bethesda, MD 20892, USA; denise.liberton@gmail.com (D.K.L.); nadine.almpani@nih.gov (K.A.); bassimc@mcmaster.ca (C.B.); 2Department of Orofacial Sciences, School of Dentistry, University of California, San Francisco, CA 94143, USA; mishrar@uw.edu; 3Metabolic Medicine Branch, National Human Genome Research Institute, National Institutes of Health, Bethesda, MD 20892, USA; carol.vanryzin@nih.gov; 4Department of Genetics and Genomic Sciences, Icahn School of Medicine at Mount Sinai, New York, NY 10029, USA; bdwebb@wisc.edu (B.D.W.); jabs.ethylin@mayo.edu (E.W.J.); 5Department of Pediatrics, University of Wisconsin School of Medicine and Public Health, Madison, WI 53705, USA; 6Department of Clinical Genomics, Mayo Clinic, Rochester, MN 55902, USA; 7Departments of Neurology and Ophthalmology, Boston Children’s Hospital, Boston, MA 02115, USA; elizabeth.engle@childrens.harvard.edu; 8Departments of Neurology and Ophthalmology, Harvard Medical School, Boston, MA 02115, USA; 9Howard Hughes Medical Institute, Chevy Chase, MD 20815, USA; 10Center for Precision Health Research, National Human Genome Research Institute, National Institutes of Health, Bethesda, MD 20892, USA; francis.collins@nih.gov

**Keywords:** facial weakness disorders, moebius syndrome, Oral Health Impact Profile, Carey Fineman Ziter syndrome, hereditary congenital facial palsy, congenital fibrosis of the extraocular muscles, smile surgery, OHIP-14, OHRQoL

## Abstract

Congenital facial weakness (CFW) encompasses a heterogenous set of rare disorders presenting with decreased facial movement from birth, secondary to impaired function of the facial musculature. The aim of the present study is to provide an analysis of subject-reported oral health-related quality of life (OHRQoL) in congenital facial weakness (CFW) disorders. Forty-four subjects with CFW and age- and sex- matched controls were enrolled in an Institutional Review Board (IRB)-approved study. Demographic data, medical and surgical history, comprehensive oral examination, and the Oral Health Impact Profile (OHIP-14) were obtained. Compared to unaffected controls, subjects with CFW had higher OHIP-14 scores overall (mean ± SD: 13.11 ± 8.11 vs. 4.46 ± 4.98, *p* < 0.0001) and within five of seven oral health domains, indicating decreased OHRQoL. Although subjects with Moebius syndrome (MBS) were noted to have higher OHIP-14 scores than those with Hereditary Congenital Facial Paresis (HCFP), there was no significant correlation in OHIP-14 score to age, sex, or specific diagnosis. An increase in OHIP-14 scores in subjects was detected in those who had undergone reanimation surgery. In conclusion, subjects with CFW had poorer OHRQoL compared to controls, and subjects with MBS had poorer OHRQoL than subjects with HCFP. This study provides better understanding of oral health care needs and quality of life in a CFW cohort and suggests that guidelines for dental treatment are required.

## 1. Introduction

Rare disorders are defined as those that affect 200,000 or fewer people in the United States. Many of these disorders affect the head, neck, and oral cavity, and are likely to have distinct effects on an individual’s Oral Health-Related Quality of Life (OHRQoL) [1]. Congenital facial weakness (CFW) encompasses a heterogenous set of rare disorders that result from a primary defect in the motor nucleus of the facial nerve or the facial nerve itself (cranial nerve 7; CN7; neurogenic), the neuromuscular junction, the muscle (myopathic), or from other unknown or mixed causes. These individuals may have “mask-like” facies with decreased ability to make facial expressions, lagophthalmos, an open mouth posture, drooling, and/or an inability to whistle and/or articular deficiencies with labial consonants [2].

Among the various forms of CFW, Moebius syndrome (MBS, MIM: 157900) is defined as congenital, non-progressive facial weakness, and limited abduction of one or both eyes due to hypoplasia or absence of the facial (CN7) and abducens (CN6) nerves [3,4,5]. Some individuals with MBS also have lower cranial nerve involvement (CN9-12) with limitations of tongue movement and velopharyngeal insufficiency. MBS occurs sporadically and its etiology is unknown; however, environmental, and genetic causes have been implicated in a subset of affected individuals [6,7,8].

The differential diagnosis of CFW also includes hereditary congenital facial paresis (HCFP) (MIM: 601471 and 604185, autosomal dominant; 614744, autosomal recessive), Carey Fineman Ziter syndrome (CFZ) (MIM: 254940, autosomal recessive), and congenital fibrosis of the extraocular muscles type 3A (CFEOM3A) (MIM: 600638, autosomal dominant) [2,9,10]. HCFP is a neurogenic isolated congenital unilateral or bilateral complete or partial facial weakness with full ocular motility. If familial, it could segregate as an autosomal dominant or recessive trait. Two dominant loci, HCFP1 and HCFP2, and one recessive locus, HCFP3, have been defined [9,11,12]. HCFP3 results from loss of *HOXB1* expression, while HCFP1 results from noncoding variants that alter *GATA2* gene expression [13].

CFZ is a myopathic congenital bilateral facial weakness, upturned/broad nasal tip, micro/retrognathia, generalized muscle hypoplasia with mild weakness, delayed motor milestone, and normal cognition. Affected individuals do not have abducens palsy but can have minimal eye movement limitations on extreme positions of gaze [13]. Affected individuals harbor homozygous or compound heterozygous mutations in myomaker (*MYMK*), a protein necessary for the fusion of muscle cells (myoblasts) into muscle fibers (myotubes) during embryonic development [14,15,16,17].

Finally, CFEOM3A is a neurogenic congenital ptosis and limitation in vertical eye movements with or without limitations in horizontal movements. It can result from variants in *TUBB3*, and specific variants result in CFEOM3 together with congenital facial palsy, lower cranial nerve dysfunction, intellectual and social disabilities, and/or a progressive sensorimotor axonal peripheral neuropathy [10,18,19].

Adults with FP often report experiencing higher levels of anxiety and depression than the general population, as well as high levels of appearance-related distress, poor social well-being, and low health-related quality of life (QOL) [20,21,22,23,24]. Given these associations, surgical reanimations involving nerve or muscle transfers have been performed in subjects with facial weakness and are thought to improve quality of life through improved oral function, facial symmetry and aesthetics, and social interactions [25,26,27,28,29,30,31,32].

The surgical reanimation surgery of choice for bilateral facial weakness is neurovascular free muscle transfer using the gracilis and pectoralis minor muscles [33]; for unilateral facial weakness, cross-facial nerve grafting is also performed [34]. An alternative is “sling surgery,” which provides muscle reattachment to the perioral muscles to prevent lower lip procumbence and drooling by suspending and supporting the lower lip [35]. These “smile surgery” methods are used to improve facial function, and they often have a positive impact on overall quality of life. The specific impact of congenital facial weakness and of reanimation surgery on OHRQoL, however, has not been reported.

A validated instrument used to assess OHRQoL is the Oral Health Impact Profile (OHIP), which is based on the World Health Organization classification of impairment, disability, and handicap. This instrument attempts to comprehensively capture functional and psychosocial outcomes of oral disorders on physical, mental, and social well-being [36,37]. OHIP is a 49-item measure divided into seven theoretical domains, including functional limitation, pain, psychological discomfort, physical disability, psychological disability, social disability, and handicap [36]. A short version of OHIP, the OHIP-14, contains 14 of the 49 items and demonstrates high reliability, validity, and precision [38]. The OHIP-14 questionnaire explores the relationship between problems with the subject’s teeth, mouth, or dentures and their daily life, documenting how often each of the 14 items occurred during the previous month. Slade and Spencer have also suggested that measures of oral health status may be used to advocate and improve understanding of how individuals perceive oral health needs and what oral health outcomes drive them to seek health care [39].

There is a need to understand the impact of rare disorders on OHRQoL, given that nearly 30 million Americans are affected by a rare disease [1]. There are very few studies that have focused on self-reported oral health in rare disorders with congenital facial weakness [40]. The purpose of this study is to investigate the impact of CFW on oral health-related quality of life by comparing OHIP-14 scores in affected subjects to a control cohort. Based on clinical experience, we hypothesized that individuals with facial weakness will have poorer OHRQoL than controls, and that individuals with facial weakness who had undergone surgical reanimation would have higher OHRQoL than those who had not.

## 2. Methods

### 2.1. Study Participants

This is a prospective cohort analysis of subjects enrolled and consented or assented onto the Institutional Review Board (IRB)-approved protocol “Study on Moebius Syndrome and Congenital Facial Weakness Disorders” (ClinicalTrials.gov Identifier: NCT02055248) at the National Institutes of Health (NIH) Clinical Center. Sixty-eight research participants with facial weakness visited the Dental Clinic of the NIH Clinical Center between 2014–2016. Their evaluation included a dental and craniofacial examination, intra- and extra-oral photos, cone-beam computed tomography (CBCT) imaging when clinically indicated, and completion of dental and oral questionnaires including the OHIP-14. Out of the 68 subjects enrolled, 44 both had a dental exam and completed the OHIP-14 questionnaire and are included in this analysis.

We first categorized the facial weakness cohort (44 individuals) into five distinct groups based on clinical and/or genetic diagnosis:Moebius syndrome (MBS): All individuals in this group were simplex cases with sporadic MBS.Hereditary congenital facial paresis (HCFP).Carey-Fineman-Ziter syndrome (CFZ).Congenital fibrosis of the extraocular muscles type 3A (CFEOM3A).Other (congenital cranial dysinnervation disorders (CCDDs) and neuromuscular (NMD) unspecified): A mixed group of other congenital disorders including CCDD or NMD not otherwise specified, that do not fit into the other four categories.

We then categorized the affected individuals based on whether they had undergone reconstructive surgery to improve lip support or facial animation (Appendix A).

Thirty-nine individuals were selected from a larger pool of unaffected family members and healthy volunteers evaluated at the Dental Clinic of the NIH Clinical Center to form the control group. Participants in this group were consented or assented onto an IRB-approved protocol (NCT02639312, PI: Lee). The control subjects were age- and sex- matched against the 44-person facial weakness cohort to the best of our ability. A detailed screening of their medical history was completed by the authors to confirm their inclusion as controls.

### 2.2. Oral Health Impact Profile

The OHIP-14 consists of 14 questions related to problems with the subject’s teeth, mouth, or dentures. Each question is assigned a value from 0 to 4 based on a frequency scale: 0 = never, 1 = hardly ever, 2 = occasionally, 3 = often, 4 = very often. These questions are divided into seven oral health domains that allow us to explore the impact of discrete aspects of the subject’s oral function on their quality of life. The seven domains include: (1) function limitations, (2) physical pain, (3) psychological discomfort, (4) physical disability, (5) psychological disability, (6) social disability, and (7) handicap. OHIP scores can range from 0 to 56 overall and from 0 to 8 within each domain and were calculated in an additive fashion. A higher score is indicative of poorer OHRQoL.

Questionnaires were completed by the participants themselves in most cases. For children less than 10 years of age (N = 6) questionnaires were completed by one of their parents, and for individuals with cognitive delay (N = 1), they were completed by their parent by proxy. Although an alternate pediatric version known as OHIP-19 is typically used in younger children, all the questions from the OHIP-14 are included in the OHIP-19, and due to the limited number of pediatric facial weakness subjects, we analyzed only OHIP-14 data. For all participants, incomplete questionnaires were excluded from the analysis.

### 2.3. Data Collection

Data collection was completed with the use of OHIP-14 questionnaire print outs that were saved as source documents in a safe location within the NIH Clinical Center. The responses of the participants were also saved to a secure online database (REDCap). Access to both source documents and the REDCap database was restricted to those involved in the data collection, curation, and analysis.

### 2.4. Statistical Analysis

Statistical tests were performed on the additive OHIP-14 scores using R 3.6.1 [41]. First, we calculated univariate statistics within each cohort including mean age, standard deviation, and range. We then calculated the mean scores and standard deviations for the overall OHRQoL score and for each of the seven domains for each cohort. We performed Shapiro–Wilk tests to test the normality of distribution and used Wilcoxon rank-sum tests to compare the OHRQoL in subjects with facial weakness to that of the control group. We applied a Bonferroni correction to counteract the effects of multiple comparisons.

Within the facial weakness cohort, we also tested for associations between OHIP-14 scores and subjects’ demographic and medical history. We used Wilcoxon rank-sum tests to identify associations between OHIP-14 scores and demographic variables including subject age (child defined as less than 18 years old, or adult defined as age 18 or older) and sex (male or female), as stated by the participants, which coincided with the sex that they were assigned with at birth. We performed Kruskal–Wallis tests to examine differences on OHRQoL among the five facial weakness diagnoses (MBS, HCFP, CFZ, CFEOM, and Other).

Additionally, given that MBS usually presents with a more severely affected craniofacial phenotype in comparison to HCFP, and that they also differ in the mode of inheritance (sporadic or familial), we compared these two diagnoses using Wilcoxon rank-sum tests to examine the effects of inheritance pattern on OHRQoL. Finally, we compared those who had or had not undergone any facial reconstructive surgery for congenital facial weakness (such as reanimation or sling surgery). Given the small sample size, corrections for multiple testing were performed for all comparisons. *p* values < 0.05 were considered statistically significant.

## 3. Results

### 3.1. Demographics

The mean age of the facial weakness cohort was 28.29 years (range: 6–64 years), while the mean age of the controls was 30.11 (range: 5–69 years). There was no significant difference in the mean age of the two cohorts (*p* = 0.626). Two female individuals who were <10 years old were age-matched to male individuals due to the lack of sex-matched controls of the same age, although there were more females in both groups. There is limited evidence regarding the pre-pubertal sex differences in dental and behavioral health [42]. The baseline demographic characteristics are presented in Table 1.

### 3.2. Subjects with CFW Compared to Controls

The overall OHIP-14 scores, as well as the domain specific scores, were found to deviate from normality according to the Shapiro–Wilk tests (*p* < 0.05 for all comparisons). Therefore, we performed non-parametric testing for all further analyses. Subjects with facial weakness had significantly higher OHIP-14 scores overall when compared to the control group (*p* < 0.0001), indicating lower OHRQoL in the affected cohort. Additionally, subjects with facial weakness had higher OHIP-14 scores for all seven oral health domains. These increases were notable compared to controls in the domains of functional limitation (*p* < 0.0001), physical pain (*p* = 0.0003), psychological discomfort (*p* < 0.0001), physical disability (*p* = 0.0360), and psychological disability (*p* < 0.0001), but not for social disability (*p* = 0.110) and handicap (*p* = 0.265). Of the five significant domains, all except physical disability remained statistically significant after multiple testing correction. A detailed summary of the mean OHIP-14 scores by cohort for each domain can be found in Table 2.

### 3.3. Comparisons within the Facial Weakness Cohort

Comparing the five diagnostic categories within the CFW cohort, we found no significant relationships between age (*p* = 0.549), sex (*p* = 0.339), inheritance pattern (*p* = 0.141) and the OHIP-14 scores. However, there was a notable increase regarding the relationship of the inheritance pattern and OHIP-14 scores between a familial form of palsy (HCFP) versus a sporadic form of palsy (MBS) (*p* = 0.008). A Kruskal–Wallis test found suggestive differences among the five diagnoses included in the study (*p* = 0.066). A detailed comparison of all results is provided in Table 3.

To further explore the differences between HCFP and MBS, we performed an additional comparison between these two diagnoses, and found lower OHIP-14 scores (better OHRQoL) in subjects diagnosed with HCFP compared to subjects diagnosed with MBS (*p* = 0.014). When we examined the seven oral health domains, we found that MBS subjects had higher OHIP-14 scores (lower OHRQoL) for each of the seven domains, and these increases were greater for functional limitation (*p* = 0.005), psychological discomfort (*p* = 0.048), and physical pain (*p* = 0.033), with suggestive increases in psychological (*p* = 0.082) and social disability (*p* = 0.064). A detailed summary of all results according to inherited status is provided in Table 4.

### 3.4. Subjects with CFW Who Had Reconstructive Facial Surgery Compared to Those Who Had Not

We found that subjects who had reconstructive facial surgery reported higher OHIP-14 scores (worse OHRQoL) than subjects who had not undergone surgery (*p* = 0.018). We then compared the OHIP-14 scores in each of the seven oral health domains to explore what contributes to this difference. We found that subjects who had undergone reanimation or sling surgery reported significantly higher OHIP-14 scores in the domains of psychological discomfort (*p* = 0.007) and psychological disability (*p* = 0.023). A detailed summary of all results according to surgical status is provided in Table 5.

## 4. Discussion

Based on OHIP-14 scores, we found a decrease in oral health-related quality of life (OHRQoL) in subjects with CFW compared to controls. Comparisons of the seven oral health-related domains indicated that functional limitation, physical pain, psychological discomfort, physical disability, and psychological disability were elevated in the facial weakness cohort compared to the controls.

We did not find differences between children and adults or between females and males, suggesting that lower OHRQoL is consistent throughout the lifetime of these subjects and does not differ between sexes. We did find a suggestive difference among the five diagnoses included in the study. This may indicate that the OHRQoL is affected more strongly in certain diagnoses than in others. However, given the small sample size of CFZ and the Other diagnoses, it is difficult to determine with certainty. In addition, MBS, CFZ, CFEOM3A, and Other are disorders that include facial weakness as well as extra-cranial physical and/or intellectual impairments that may also play a role in determining OHRQoL.

Our results comparing the OHRQoL between MBS and HCFP cohorts are intriguing because they do have a different severity of the degree of craniofacial differences, but they also differ in terms of inheritance pattern. Affected individuals in the HCFP cohort in our study have similarly affected family members present in their life. In our cohort, MBS is a sporadic disorder with no known genetic cause, and these subjects are the only individual in their family with a craniofacial anomaly. According to our analysis, the HCFP cohort had significantly lower OHIP-14 scores overall, indicating better OHRQoL than the Moebius cohort, particularly in the domains of functional limitation and psychological discomfort. It is, therefore, possible that the better OHRQoL in the HCFP cohort is a direct result of the less severe phenotype, but there could be an additional impact of the reduced isolation, and reduced psychological burden in individuals who are part of an affected family network. Further research is needed to confirm the cause of this difference and the potential role of family support in OHRQoL.

Interestingly, contrary to previous studies that have indicated that facial surgery improved overall quality of life [29,43], the 10 subjects who had facial reanimation or sling surgeries did not report better OHRQoL compared to those who had not undergone any surgery, particularly in the domains of psychological discomfort and psychological disability. Of the subjects who had undergone surgery, our clinical examinations demonstrated limited facial function and animation in all but one subject, as well as limited mouth opening because of significant scar tissue. Additionally, most of the subjects who had facial surgery had notably worse oral hygiene, which could be associated with the restricted mouth opening. Each of these limitations can lead to further deterioration in their oral health status, worse OHRQoL, and thus increased psychological discomfort. Given the fact that the OHRQoL scores of subjects who had undergone facial reanimation surgery, were lower than the scores of subjects that had not undergone surgery, more investigation is required to determine the effects and value of this type of facial surgery on OHRQoL, facial function, and oral health status. A comparative study to evaluate the pre- and post-questionnaires in the same individuals would also assist in the assessment of the impact of facial reanimation surgery in OHRQoL.

One of the limitations of the study is the relatively small sample size. While a significant obstacle, this should not deter the progress of research in rare disorders. Additionally, comparison and stratification based on different diagnostic groups are difficult when the total sample size is small. However, based on our findings, there is some variability among the different types of CFW and robust social support systems, and strong family networks may improve OHRQoL by reducing social isolation. Finally, further studies are needed to assess the clinical impact of the differences reported in this cohort. The reason is that the Minimal Important Difference (MID) in the case of OHIP-14 has only been validated in studies examining the impact of a certain intervention, comparing pre- and post-intervention scores [44,45,46]. However, there is no study to date providing an MID score in comparative studies.

These results support the hypothesis that individuals with congenital facial weakness have a decreased OHRQoL in comparison to healthy individuals. These aspects have received little attention in the past, as these disorders are rarely life threatening, and the oral cavity has historically been dissociated from the rest of the body when considering general health status [47,48,49]. Recent research has begun to highlight that oral disorders have emotional and psychosocial consequences as serious as other disorders [50]. The face is an important component in interpersonal contact and is important for establishing rapport, social acceptance, and gainful employment [47]. However, the hallmark of these CFW conditions is lack of facial expression, which may be misinterpreted as reflecting irritation, lack of interest, or boredom [40]. Other oral features that were previously mentioned, like drooling and potential speech impairments and difficulty eating, can adversely affect social interactions and, therefore, lead to further isolation and limited participation in activities [47,50]. It is becoming increasingly obvious that oral health is not a separate domain from overall health and can have important consequences for health outcomes [37,48]. Furthermore, subjects’ assessments of their own health-related quality of life can differ from the opinion of health care professionals. Especially in the case of rare diseases, the need for a more detailed assessment of the patient’s perception is required to better understand the impact of the craniofacial differences and functional disparities in the quality of life of these individuals. These conditions, due to their unique nature, do not allow for an objective insight from the clinicians. The OHIP-14 questionnaire is a validated widely used tool that could be used for this purpose.

## 5. Conclusions

A decreased OHRQoL was detected in patients with CFW in comparison to healthy controls. Individuals with a clinical diagnosis of MBS reported lower OHRQoL in comparison to those with an HCFP diagnosis, which could be associated with differences in the severity of the craniofacial phenotype and their inheritance pattern. This information can help guide good clinical practices aimed at improving oral health individuals affected by CFW disorders.

## Figures and Tables

**Table 1 ijerph-21-00615-t001:** Demographic information on study participants.

Variable		Facial Weakness Cohort	Control Cohort
**Total Subjects (N)**		44	39
**Gender**	**Male: N (%)**	15 (34.1%)	17 (43.6%)
	**Female: N (%)**	29 (65.9%)	22 (56.4%)
**Age**	**Mean (Years)**	28.29	30.11
	**SD**	17.01	16.86
	**Range**	6–64	5–69
**Diagnosis**	**MBS (N)**	22 (50%)	N/A *
	**HCFP (N)**	9 (20.5%)	
	**CFZ (N)**	3 (6.8%)	
	**CFEOM3A (N)**	7 (15.9%)	
	**Other (N)**	3 (6.8%)	

* *Non-applicable*.

**Table 2 ijerph-21-00615-t002:** Comparison of mean and standard deviation of OHIP-14 scores in the facial weakness and control cohorts within each of the seven oral health domains and overall.

Dimension	Facial Weakness	Control	*p*-Value
	Mean (SD)	Mean (SD)	
Functional Limitation	2.59 (1.56)	0.25 (0.67)	<0.0001
Physical Pain	2.02 (1.81)	0.64 (1.06)	<0.0001
Psychological Discomfort	3.20 (2.11)	1.33 (1.91)	<0.0001
Physical Disability	1.34 (1.53)	0.59 (1.04)	0.036 *
Psychological Disability	2.18 (1.53)	0.85 (1.31)	<0.0001
Social Disability	1.18 (1.57)	0.62 (0.96)	0.110
Handicap	0.59 (1.20)	0.17 (0.39)	0.264
**Total OHIP**	**13.11 (8.11)**	**4.46 (4.98)**	**<0.0001**

*p*-values were obtained using Wilcoxon rank-sum tests. The total OHIP-14 scores in each cohort are in bold.

**Table 3 ijerph-21-00615-t003:** Comparison of OHIP-14 scores within the facial weakness cohort and associations with demographic or medical history.

Variables		N (%)	Additive (Total OHIP)	
			Mean (SD)	*p*-Value
**Age**	**Child (<18 years)**	16	12.06 (8.20)	0.549
	**Adult (≥18 years)**	28	13.71 (8.16)	
**Sex**	**Male**	15	11.73 (9.43)	0.339
	**Female**	29	13.83 (7.42)	
**Diagnosis**	**MBS**	22	16.00 (8.11)	0.066
	**HCFP**	9	7.22 (6.74)	
	**CFZ**	3	15.33 (5.03)	
	**CFEOM3A**	7	10.86 (7.81)	
	**Other**	3	12.67 (7.77)	
**Inheritance Pattern**	**MBS (Sporadic)** **HCFP (Familial)**	22 9	16.00 (8.11) 7.22 (6.74)	0.008
**Surgery**	**Yes**	10	17.70 (4.21)	0.018
	**No**	34	11.76 (8.53)	

*p*-values were obtained using Wilcoxon rank-sum tests or Kruskal–Wallis tests.

**Table 4 ijerph-21-00615-t004:** Comparison of mean and standard deviation of OHIP-14 scores in the subjects who had HCFP (familial) compared to subjects with Moebius syndrome (sporadic) within each of the seven oral health domains and overall.

Dimension	HCFP, N = 9	MBS, N = 22	*p*-Value
	Mean (SD)	Mean (SD)	
Functional Limitation	1.22 (1.20)	2.86 (1.17)	0.005
Physical Pain	0.78 (1.64)	2.50 (1.92)	0.033
Psychological Discomfort	2.00 (2.06)	3.77 (2.11)	0.048
Physical Disability	1.00 (1.58)	1.55 (1.97)	0.486
Psychological Disability	1.44 (1.59)	2.68 (1.46)	0.082
Social Disability	0.44 (1.01)	1.55 (1.74)	0.064
Handicap	0.33 (0.71)	0.73 (1.39)	0.711
**Total OHIP**	**7.22 (6.74)**	**16.00 (8.12)**	**0.008**

*p*-values were obtained using Wilcoxon rank-sum tests. The total OHIP-14 scores in each cohort are in bold.

**Table 5 ijerph-21-00615-t005:** Comparison of mean and standard deviation of OHIP-14 scores in the subjects who had facial surgery and those who did not within each of the seven oral health domains and overall.

Dimension	Had Surgery, N = 10	No Surgery, N = 34	*p*-Value
	Mean (SD)	Mean (SD)	
Functional Limitation	3.33 (1.06)	2.38 (1.63)	0.100
Physical Pain	2.70 (1.83)	1.82 (1.78)	0.164
Psychological Discomfort	4.70 (1.49)	2.76 (2.09)	0.007
Physical Disability	1.80 (1.99)	1.21 (1.65)	0.355
Psychological Disability	3.00 (0.94)	1.94 (1.59)	0.023
Social Disability	1.60 (1.26)	1.06 (1.65)	0.092
Handicap	0.60 (0.84)	0.58 (1.31)	0.376
**Total OHIP**	**17.70 (4.22)**	**11.76 (8.53)**	**0.018**

*p*-values were obtained using a Wilcoxon rank-sum. The total OHIP-14 scores in each cohort are in bold.

## Data Availability

The datasets used and analyzed during the current study are available from the corresponding author on reasonable request.

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
