# Peer review of "Oral Health-Related Quality of Life in Rare Disorders of Congenital Facial Weakness"

_ijerph, 2024, doi:10.3390/ijerph21050615_

Round 1

Reviewer 1 Report

Comments and Suggestions for Authors

This article contributes to the understanding of the problems faced by people with rare diseases. In particular, Oral Health Related Quality of Life (OHRQoL) is often neglected and has received some attention as a result of this scientific work. It is not surprising that OHRQoL is worse than in a healthy comparison group, but studies like this are still needed to quantify and assess this.

The manuscript covers a topic of great interest and is written with skill. The methodology is generally well presented and the conclusions are easy to understand.

Abbreviations:

-          It is recommended to provide explanations for all abbreviations when they are first used. Examples include line 29 with the abbreviation 'IRB', which has not been defined yet, and line 34 with 'HCFP'.

References:

-          It may be worth considering reviewing the citations throughout the text as there may be some inaccuracies. It seems that there may have been an error in the assignment of references, as reference number 29 onwards may not be correctly labelled.

-          It would also be advisable to provide the original source in line 44 (à Orphan Drug Act).

-          It may be worth noting that the first names in the reference number 11 in lines 463 to 465 are spelled out, which appears to be inconsistent with the rest of the references. It could be suggested that this be corrected for the sake of consistency.

Material and Methods

-          There is a lack of information on the information and consent of participants or their legal representatives.

-          There is also a lack of information about data protection and how it has been and will be guaranteed.

-          Lines 107-111: Unfortunately, the dental examinations and questionnaires mentioned are not specified and explained in more detail. Perhaps further information can be provided on this?

-          For sub-point 2.2. I would recommend explaining why OHIP14 was used, even if there are other tools for the age, and not just in the discussion. I find the reasoning understandable, but I was looking for it at this point.

-          For subsection 2.3, I would like to know what value is considered significant. Usually a p-value less than the significance level α = 0.05 is considered statistically significant and I think this is the case for the results presented here.

Results

-          I would be very interested in the results of the clinical examinations and the other questionnaire results mentioned in the methods section. Are there correlations with OHRQoL in addition to the underlying rare disorder?

-          My recommendation for Table 1 would be an additional column for the p-value.

-          I also find 3 decimal places for p-values to be sufficient and easier to read. So <0.001 would often be sufficient. This gives the text a more uniform appearance and makes it easier to understand the meaning of the figures.

-          Line 244: Is the p-value given here correct? It is not in the table. There it is 0.008.

-          Line 247: Is the p-value for physical pain (p=0.033) not significant?

Discussion

-          Line 305: Where are the results of the clinical examination?

-          Line 308: What index was used to measure oral hygiene and what was the result?

-          Line 312: Were the OHRQoL scores higher and the corresponding quality of life lower? Or am I misunderstanding something at this point?

Author Response

Please find attached the responses to the comments for Reviewer 1 in the attached word document. 

Reviewer 2 Report

Comments and Suggestions for Authors

It is a cross-sectional study that aimed to provide an analysis of subject-reported oral health-related quality of life (OHRQoL) in congenital facial weakness disorders in 44 patients.

These are the necessary coercions:

1. Abstract: "Compared to unaffected controls, subjects with CFW had higher OHIP-14 scores overall and within five of seven oral health domains, indicating decreased OHRQoL." - what are the differences and are they statistically significant? Please correct the sentence.

2. Abstract: "..scores than those with HCFP.." - what does this abbreviation mean?

3. Abstract: please write as specific a conclusion as possible with regard to the obtained results.

4. Methodology. Please transfer the parts irrelevant to the methodology from lines 112 to 139 (etiology, description of the condition) to the introduction, and here you only find the conditions that you took as the test group.

5. Methodology: How was the sample size or research power calculated? Where was the research conducted, when? Please meet CONSORT guidelines!!?

6. Methodology: which statistical package and which tests? Who ethically approved the research? Informed consent?

7. Results: in TBl 1, next to the n in the diagnosis, put %!

8. Results: Table 2 unclear, what are those p values? Put the P value in the test to three decimal places.

9. Place some parts from the table titles at the bottom of them.

10. Results: in table 4, put the results of OHIP dimensions for all observed conditions and, depending on the distribution of the results, check with parametric or non-parametric test. the tables are not well arranged and take up a lot of space, align them.

11. Discussion: The discussion is very short, please explain the obtained results better. As far as I can see in my experience, there are NO REFERENCES with which you compared the results, while they exist in the conclusion?!

11. Conclusion: Too general, please shorten it without references, base the conclusion on the most important results.

Author Response

Please find attached the responses to the comments for Reviewer 2. 

Round 2

Reviewer 2 Report

Comments and Suggestions for Authors

The abstract is still inadequately designed, figures and results are missing. These results are insufficiently representative in the scope of the abstract. Who cares in the abstract that the research was approved by the ethics committee.

What statistical tests were performed on the additive OHIP-14 scores using R 3.6.1 [41]?

The manuscript is messy and hard to read, please edit according to the instructions.

Comments on the Quality of English Language

English is incomprehensible, needs additional checking.
